# Increasing the Activity of the High-Fidelity SpyCas9 Form in Yeast by Directed Mutagenesis of the PAM-Interacting Domain

**DOI:** 10.3390/ijms25010444

**Published:** 2023-12-28

**Authors:** Artem I. Davletshin, Anna A. Matveeva, Stanislav S. Bachurin, Dmitry S. Karpov, David G. Garbuz

**Affiliations:** 1Engelhardt Institute of Molecular Biology, Russian Academy of Sciences, 119991 Moscow, Russia; artem.dav7@yandex.ru (A.I.D.); wooflit@gmail.com (A.A.M.); aleom@yandex.ru (D.S.K.); 2FSBEI HE Rostov State Medical University, Ministry of Health, 344022 Rostov-on-Don, Russia; bachurin.rostgmu@gmail.com; 3Center for Precision Genome Editing and Genetic Technologies for Biomedicine, Engelhardt Institute of Molecular Biology, Russian Academy of Sciences, 119991 Moscow, Russia

**Keywords:** *Saccharomyces cerevisiae*, genome editing, site-directed mutagenesis, CRISPR/Cas9, high-fidelity SpyCas9 forms

## Abstract

CRISPR/Cas systems are used for genome editing, both in basic science and in biotechnology. However, CRISPR/Cas editors have several limitations, including insufficient specificity leading to “off-targets” and the dependence of activity on chromatin state. A number of highly specific Cas9 variants have now been obtained, but most of them are characterized by reduced activity on eukaryotic chromatin. We identified a spatial cluster of amino acid residues in the PAM-recognizing domain of *Streptococcus pyogenes* Cas9, whose mutations restore the activity of one of the highly specific forms of SpyCas9 without reducing its activity in *Saccharomyces cerevisiae*. In addition, one of these new mutations also increases the efficiency of SpyCas9-mediated editing of a site localized on the stable nucleosome. The improved Cas9 variants we obtained, which are capable of editing hard-to-reach regions of the yeast genome, may help in both basic research and yeast biotechnological applications.

## 1. Introduction

Clustered regularly interspaced short palindromic repeats-associated protein (CRISPR/Cas) is an adaptive response system against foreign DNA in prokaryotes and consists of repeated sequences separated by unique spacers. Currently, these systems are used to introduce targeted mutations into the genomic DNA (genome editing) of prokaryotes and eukaryotes in fundamental science, biotechnology, and biomedicine applications. The CRISPR/SpyCas9 system from *Streptococcus pyogenes* is most commonly used because of its high activity and simplicity [1]. On the other hand, a serious disadvantage of this system is the relatively high off-target activity (the ability to cause mutations in genome regions partially matching the sgRNA) and the high dependence of the editing efficiency on the chromatin structure (in areas of densely packed chromatin, the editing efficiency is low) [2,3,4,5]. These problems limit the use of the CRISPR/SpyCas9 system in a variety of fields, including basic yeast research and biotechnology [6,7].

Briefly, genome editing technology uses chimeric single guide RNAs (sgRNAs), which consist of a structural part recognized by the SpyCas9 endonuclease and a genome-specific part (spacer) 20 nucleotides long that is complementary to the target sequence in the genome (protospacer). The sgRNA/Cas9 complex binds to the specific site of the genome DNA, and SpyCas9 endonuclease introduces a double-stranded DNA break (DSB) into the genomic target. Further, at the site of the break, a DNA fragment flanked by sequences homologous to the sites surrounding the break can be inserted due to the mechanism of DSB repair, mainly through homologous recombination in the case of yeast [8,9]. In other organisms, such as mammalian cells, homologous recombination is inefficient, and DSBs are repaired predominantly through the non-homologous end joining mutagenic pathway [8]. The most commonly used genome editing system, CRISPR/SpyCas9, requires the presence of a protospacer adjacent motif (PAM) with an NGG sequence adjacent to the target site for DNA hydrolyzing [10]. sgRNA/Cas9 binds primarily to the PAM using the PAM-interacting domain (PID), and only after that does the interaction of sgRNA with the target DNA and DNA cleavage occur [11,12]. Accordingly, it is possible to search for mutations within the PID that can increase SpyCas9 activity.

The ability of SpyCas9 to hydrolyze DNA with incomplete complementarity between the sgRNA spacer and the protospacer causes non-targeted DNA breaks (off-targets) [13]. The ability of SpyCas9 to recognize PAM with sequences of NAG and NGA also increases the number of potential off-target sites [14]. One of the main directions for improving the SpyCas9 system is the introduction of amino acid substitutions that increase the specificity of SpyCas9 with respect to protospacer and PAM [15,16]. Unfortunately, mutations that increase the specificity of SpyCas9 lead to a decrease in activity (percentage of edited target DNA sites) [17].

Mutations that increase SpyCas9 activity for on-target sites are also described [18,19]. In the previous work, we combined individual mutations that increase specificity with mutations that increase SpyCas9 activity and obtained several SpyCas9 variants with high specificity and maintaining high activity close to that of the wild-type SpyCas9 in the yeast *Saccharomyces cerevisiae* [19]. We also obtained a new mutation, L1206P, within the PID that restores the on-target activity of a number of high-fidelity SpyCas9 forms and increases the activity of wild-type SpyCas9 on the “difficult” site with stable nucleosomes in *S. cerevisiae* [19]. In the present work, we rationally mutated the PID of SpyCas9 in order to (1) restore the activity of high-fidelity SpyCas9 form SniperDE while maintaining its high specificity and (2) facilitate the editing of DNA in the context of yeast chromatin. This result may be useful for more accurate and efficient editing of yeast genomes in basic science and biotechnology.

## 2. Results

### 2.1. Design of New Mutations within SpyCas9 PID with Potential Gain-of-Function Effect

Previously, we obtained an L1206P substitution in PID that increased the on-target activity of some highly specific forms of SpyCas9 in *S. cerevisiae*. We hypothesize that the mechanism of action of the L1206P substitution on SpyCas9 activity is based on a change in the position of two arginine residues (R1333 and R1335) that play a key role in PAM recognition [19]. Presumably, the L1206P substitution compensates for the effect of the D1135E mutation, which increases PAM specificity but decreases SpyCas9 activity [19]. Now, we visualize the position of all amino acid residues around L1206 using the crystal structure of the complex SpyCas9:sgRNA:PAM-containing DNA target [20] in order to search for other suitable residues whose replacement could theoretically affect the tertiary structure of PID and enhance SpyCas9 activity. We selected two positions, E1341 and A1345, which are spatially close to L1206 and R1333/R1335 within the tertiary structure of SpyCas9 (Figure 1). We designed substitution E1341D as a replacement by a similarly charged but smaller residue, E1341H with a change in charge to positive with theoretically maximum possible influence on the local structure of the protein, A1345L (replacement with an aliphatic residue with greater volume capable of shifting the spatial arrangement of the R1333/R1335 group), and A1345P similar to L1206P, the improving mutation obtained previously [19]. The analogy is the replacement of a relatively free in-mobility alanine with a proline residue, increasing the rigidity of the polypeptide chain site and consequently changing the structure of the PAM-binding domain, similar to the L1206P substitution. Further, the effect of the designed substitutions on the SpyCas9 properties was experimentally verified.

### 2.2. Mutations E1341H and A1345L Increase the Activity of High-Specific SpyCas9 Forms SniperCas DE and iSniperCas DE, but Do Not Affect the Activity of Wild-Type SpyCas9 When Using the Most Active Spacer against ADE2 Gene

Substitutions E1341D, E1341H, A1345L, and A1345P were obtained separately and in combination with each other. We introduced these substitutions into the wild-type SpyCas9 and into the high-specific SpyCas9 forms: iSniperCas (F539S, M763I, K890N, D147Y, and P411T), SniperCas DE (F539S, M763I, K890N, and D1135E), and iSniperCas DE (F539S, M763I, K890N, D147Y, P411T, and D1135E). Also, mutations E1341D, E1341H, A1345L, and A1345P were combined with the substitution L1206P, obtained earlier, increasing the activity of high-fidelity forms of SpyCas9 in yeast [19]. The resulting variants are given in Table 1. The activity of the new mutant SpyCas9 forms was checked using the yeast *ADE2* test system described in [19] with the optimal spacer against the *ADE2* gene designated here as *ADE2*-lit [21]. This system allows rapid assessment of Cas9 activity by counting the number of edited yeast colonies by their color (see [19] and the Section 4 “Materials and Methods” for details).

Initially, to assess the effect of mutations E1341D, E1341H, A1345L, and A1345P on the activity of SpyCas9, a highly specific form of SniperCas DE, characterized by high accuracy but reduced activity compared to wild-type SpyCas9 [19,22], was used. Mutations E1341H and A1345L increase the activity of the highly specific form of SniperCas DE by two times, while the combination of both mutations does not have a synergistic effect (Figure 2A). Substitutions E1341D and A1345P, in the case of the SniperCas DE, turn out to be neutral (Figure 2A). Thus, our hypothesis of a possible beneficial effect of replacing alanine with proline at position 1345, similar to the increase in SpyCas9 activity resulting from L1206P substitution, was not confirmed. In combination with the L1206P mutation, substitutions E1341D and A1345L do not have a significant effect on the activity of the variant, mutation E1341H reduces the activity of the SniperCas DE LP variant by half, and the combination of substitutions L1206P and A1345P completely inactivates SniperCas DE (Figure 2A).

Earlier, we introduced the SniperCas variant [22] mutations D147Y and P411T described by [18]. The resulting variant, iSniperCas, has increased activity but decreased specificity relative to the original SniperCas [19]. When introduced into the iSniperCas variant, the E1341H mutation and the combination of E1341H and A1345L mutations do not affect its activity, which is already quite high compared to the original SniperCas variant (Figure 2). A combination of the original SniperCas variant with mutations D147Y, P411T, and D1135E (iSniperCas DE) was also previously obtained to produce SpyCas9 endonuclease with an optimal balance of specificity and activity [19]. The E1341H mutation and the E1341H/A1345L combination were also introduced into this variant. The E1341H substitution alone did not significantly affect iSniperCas DE activity, whereas the combination of E1341H and A1345L slightly increased iSniperCas DE activity (Figure 2A).

The introduction of substitutions E1341H and A1345L into wild-type SpyCas9 alone and in combination does not significantly increase the activity of the mutant variant relative to the original one. Notably, other mutations that increase the activity of highly specific forms of SpyCas9 had no effect on wt SpyCas9 in *S. cerevisiae* [19]. Thus, among the obtained variants, the E1341H and A1345L mutations had an effect only when combined with the D1135E substitution using the *ADE2*-lit spacer (Figure 2A).

Because we used a haploid strain of *S. cerevisiae* and *S. cerevisiae* has a very weak NHEJ repair system, off-target editing results in chromosomal breaks that cannot be efficiently repaired by highly active homologous recombination in the absence of an appropriate template. Repair only works on the edited *ADE2* locus to which the template is introduced in the form of a PCR fragment (see Section 4 Materials and Methods). We have previously shown that the number of transformed colonies grown using the *S. cerevisiae ADE2* test system is in inverse dependence on the activity of the SpyCas9 variants checked [19]. This observation holds true in the current experiments as well: fewer colonies grow after yeast transformation with SpyCas9 variants with higher activity than when transformed with variants with low activity because more active SpyCas9 forms cause more unrepaired chromosome breaks (Figure 2B). However, this estimate is too approximate, so we used experiments with the imperfect spacer containing one substitution compared with *ADE2*-lit to test the specificity of the SpyCas9 mutants we obtained.

### 2.3. The E1341H and A1345L Mutations Do Not Affect the Specificity of SpyCas9 When an Imperfect Spacer Is Used

To test the specificity of the mutant variants in the test system based on the *ADE2* gene, an imperfect spacer that differed from the *ADE2*-lit spacer by replacing G with A at the fifth position from the 5′-end was used. The SniperCas DE EH, SniperCas DE AL, and SniperCas DE EH/AL variants were chosen for testing because they showed the most significant increase in activity after the introduction of the E1341H and A1345L mutations into the original SniperCas DE variant. The specificity of wild-type SpyCas9 was also evaluated after the introduction of the E1341H and A1345L mutations. It was shown that neither individually nor jointly did the E1341H and A1345L mutations reduce the specificity of SniperCas DE and wild-type SpyCas9 towards the spacer (Figure 3A).

### 2.4. E1341H and A1345L Mutations Have a Weak Effect on the PAM Specificity of SpyCas9

Presumably, the E1341H and A1345L mutations could affect the PAM specificity of SpyCas9 by virtue of localization within the PID. To test the PAM specificity of the mutant variants used, spacers differed from the *ADE2*-lit spacer by a shift with one nucleotide upstream or downstream so that the adjacent PAM would change from NGG to NAG or NGA [19]. The E1341H and A1345L mutations do not affect the specificity of the SniperCas DE variant toward the NAG PAM (Figure 3B). In the case of wild-type SpyCas9, the A1345L mutation and the combination of the E1341H and A1345L mutations weakly increase the probability of editing by NAG PAM (Figure 3B). The frequency of editing in this case does not exceed 1.2 to 1.3%. In the case of NGA PAM, the E1341H and A1345L mutations significantly increase the frequency of editing only in combination with each other but not individually (Figure 3B).

### 2.5. The E1341H and A1345L Mutations Significantly Increase Activity of the Wild-Type SpyCas9 on Protospacers within Stable Nucleosomes

One of the obstacles to the use of CRISPR/Cas-based editing systems is their low activity towards eukaryotic chromatin. It is known that most high-fidelity forms of SpyCas9 are characterized by weak binding to DNA compared to wild-type SpyCas9, and therefore they are unable to efficiently target DNA in stable nucleosomes [4]. We investigated the effects of the E1341H and A1345L mutations on the activity of SpyCas9 against yeast nucleosomal DNA. For this purpose, we used spacers to the *ADE2* gene, designated gRNA-2(2) and gRNA-2(3), which direct SpyCas9 to DNA sites within nucleosomes with different degrees of stability, designed based on nucleosome position mapping in the *S. cerevisiae* genome [23]. The 2(3) spacer is located within the most stable nucleosome, while the 2(2) spacer is located at the edge of the same nucleosome [19]. The previously used *ADE2*-lit spacer recognizes a DNA site on the weak nucleosome so that all SpyCas9 modifications show maximum activity when it is used [19]. As can be seen in Figure 4, the high-fidelity SniperCas DE variant shows 50-fold less activity than wild-type SpyCas9 when using gRNA-2(2) and virtually no activity when using gRNA-2(3). Spacer 2(3) is the most difficult, as even wild-type SpyCas9 shows five-fold lower editing efficiency when using this guide RNA compared to the optimal *ADE2*-lit spacer (Figure 4). The A1345L mutation increases SniperCas DE activity by five- to six-fold with spacer 2(2) but does not affect activity with spacer 2(3) (Figure 4). The addition of the EH mutation has no significant effect on either alternative spacer. Also, the A1345L mutation significantly increases wild-type SpyCas9 activity with spacer 2(3), almost to the level of activity with the *ADE2*-lit spacer (Figure 4). The same effect is exerted by the L1206P mutation, also localized in PID [19]. The resulting mutations that increase SpyCas9 activity in hard-to-hydrolyze nucleosomal DNA confirm the importance of PID as a target for modifications that increase SpyCas9 activity.

### 2.6. Molecular Dynamics Modeling Suggests a Molecular Mechanism for the Increased Activity of SpyCas9 Variants with EH and AL Mutations

To find out the possible mechanisms of action of the EH and AL mutations, we performed molecular dynamics (MD) simulations on the wild-type SpyCas9 and its derivatives, as indicated in Table 2.

The results of the MD simulations (Figure 5) show that the overall geometry of SniperCas variants is not significantly changed relative to the wild-type SpyCas9. The exception is the region of 1017–1040 aa residing within the RuvCIII (924–1098 aa) catalytic domain, where the maximum spatial structural mismatch is observed.

To assess the conformational stability of the SpyCas9 variants, three root-mean-square deviation (RMSD) analyses were conducted: one for the entire SpyCas9 proteins and two for the active centers (HNH and RuvC) (Figure 6). The sharp rise followed by a rapid plateau represents a typical successfully executed MD simulation. An RMSD value of 0.45 signifies that the structure of the SpyCas9 protein remains relatively unchanged over the studied time period (Figure 6A). This stability is expected for a protein characterized by a well-defined tertiary structure consisting of numerous ordered domains. Thus, the data suggest that the mutations did not significantly affect the overall structure of SpyCas9.

Next, we examined the conformational lability of the active sites (Figure 6B,C) of the enzyme since the formation of enzyme–substrate transition states is crucial for the DNA-hydrolyzing activity of SpyCas9. Unexpectedly, we found that all mutants exhibit high RMSD values compared to wild-type SpyCas9, indicating significant mobility of amino acid residues in the active centers of SpyCas9 variants.

Since the RMSD analysis of the amino acid residues of the catalytic domains did not reveal specific features of SpyCas9 variants, we performed an RMSD analysis of the position of Mg^2+^ ions in the active centers (Figure 7A). Our results show that the largest deviation of Mg^2+^ ions is observed in the case of the SniperCas variant. The SniperCas-AL variant shows a slightly smaller deviation. The other variants do not show significant deviations from the wild type. We then analyzed the relative positions of Mg^2+^ ions in the active sites of SpyCas9 variants at the end of the MD simulation (Figure 7B). Our results show that the Sniper-EH/Sniper-DE-EH pair at the HNH center and the Sniper-EH/Sniper-DE-EH and Sniper-AL/Sniper-DE-AL pairs at the RuvC center exhibit significantly larger relative Mg^2+^ ion displacement within the pair compared to other variant pairs. This agrees well with the results of our experiments showing that the addition of the AL or AH mutation strongly induces the activity of the SniperCas-DE variant. In addition, we examined the interaction of amino acid residues in SpyCas9 active sites with Mg^2+^ ions (Figure 7C). We found no change in the coordination of amino acid residues with Mg^2+^ in the active site of HNH in all SpyCas9 variants. In the RuvC domain, only in the case of the SniperCas-EH variant did one of the Mg^2+^ ions form an additional coordination bond with asparagine at position 990. In the other variants, the conformation of the RuvC domain does not allow for efficient proximity to asparagine 990 during the modeling period.

## 3. Discussion

We have shown the presence of two new positions within the PID SpyCas9 substitutions that can increase on-target activity without loss of specificity on the same DNA locus in *S. cerevisiae*. Overall, many SpyCas9 modifications with substitutions that increase the on-target activity or specificity of the modified variants have now been obtained [24,25]. Most mutations have opposite effects on activity and specificity: modifications that increase SpyCas9 activity decrease specificity, and vice versa [19,26]. Thus, it seems promising to use combinations of substitutions, some of which increase the activity and others that increase the specificity of SpyCas9, to compensate for each other. Initially, modification of the PAM-interacting domain to improve the properties of SpyCas9 started with the D1135E substitution, which increases PAM specificity but decreases the activity of SpyCas9 [27]. Next, we obtained the L1106P substitution, which compensated for the decrease in activity upon introduction of the D1135E mutation but did not affect the specificity of mutant forms of SpyCas9 [19]. Using visualization of the SpyCas9 3D structure, we found that the L1206P mutation is spatially located near the PAM-recognizing residues (R1333, R1335) and, apparently, compensates for the effect of the D1135E substitution. The new mutations we obtained in this study, E1341H and A1345L, have the same effect. Like the L1206P substitution, they increase the activity of the highly specific form of SniperCas DE but have no or little effect on the activity of wild-type SpyCas9 and other highly active forms of SpyCas9 with the spacer matched to open chromatin. When using a spacer to the DNA sequence within a stable nucleosome, the L1206P and A1345L substitutions have no or little effect on editing efficiency in the case of highly specific SpyCas9 variants. However, when introduced into wild-type SpyCas9, the L1206P and A1345L substitutions multiply, increasing its activity against the “difficult” DNA region. It can be assumed that mutations E1341H and A1345L, as well as L1206P, affect the conformation of PID in such a way that allows SpyCas9 to win the competition with histones for DNA binding at the PAM site. The change in PID conformation is indirectly confirmed by the change in activity of mutant SpyCas9 variants on DNA targets with alternative PAMs (Figure 3). PAM binding is known to be a critical step in SpyCas9 target recognition [28,29]. We hypothesized that more efficient PAM binding in the context of chromatin may contribute to the increased activity of SpyCas9 variants with PID mutations.

Since the PID domain is spatially close to the catalytic domains, we tested whether PID mutations could affect their structural characteristics using MD simulations. Our results indicate that PID mutations did not significantly change the overall conformation of the protein or the structure of its catalytic domains. The differences are related to the mobility and position of Mg^2+^ ions and amino acid residues involved in the coordination of Mg^2+^ ions. Changes were found in the active sites of the most active mutants, SniperCas-AL and SniperCas-EH. Thus, it cannot be excluded that PID mutations can stimulate DNA hydrolysis by changing the structure of SpyCas9 active sites.

Thus, we obtained two new forms of SpyCas9 (SniperCas DE AL and SniperCas DE EH) with balanced activity and specificity (using a spacer guide tailored to open chromatin regions). In addition, the A1345L substitution can be used to modify wild-type SpyCas9 to edit DNA loci located within stable nucleosomes. The improved SpyCas9 variants should further stimulate the application of the CRISPR/Cas9 system in both basic research and biotechnological applications of the yeast *S. cerevisiae*.

## 4. Materials and Methods

### 4.1. Yeast and Bacterial Strains

We used the *S. cerevisiae* yeast strain BY4741 trp1-Δ (BY4741 trp1::URA3 [19]) and *Escherichia coli* XL1Blue chemocompetent cells (*recA1 endA1 gyrA96 thi-1 hsdR17 supE44 relA1 lac* [F′ proAB lacIqZ∆M15 Tn10 (Tetr)]).

### 4.2. Bioinformatic and Statistical Analysis

The SpyCas9 structures (PDB IDs 4oo8 [30] and 6o0y [31]) were visualized using the PyMOL Molecular Graphics System (version 2.0, Schrödinger, LLC, New York, NY, USA). The Kolmogorov–Smirnov test (https://www.socscistatistics.com/tests/kolmogorov/default.aspx, accessed 22 December 2023) was used to assess whether the data were normally distributed. Fisher’s test (https://www.statskingdom.com/220VarF2.html, accessed 22 December 2023) was used to assess whether the compared samples have equal variance. In cases where the samples compared were normally distributed and had equal variance, Student’s *t*-test (https://www.socscistatistics.com/tests/studentttest/, accessed 22 December 2023) was used to compare the mean. In cases where the compared samples were normally distributed but had different variances, Welch’s *t*-test (https://www.statskingdom.com/150MeanT2uneq.html, accessed 22 December 2023) was used to compare the mean.

### 4.3. SpyCas9 Mutagenesis and Assembly of SpyCas9 Expressing Plasmids

To assemble CRISPR-Cas9 plasmids from multiple parts, we used a modified protocol for plasmid assembly in *S. cerevisiae* by homologous recombination [19,32]. Briefly, plasmid p414-TEF1p-Cas9-CYC1t was used as a scaffold to assemble Cas9-expressing plasmids. PCR fragments of the SpyCas9 gene with the desired combination of mutations were amplified from corresponding plasmids. The oligonucleotides used as primers in PCR reactions are given in Appendix A. Purified PCR fragments were mixed with p414-TEF1p-Cas9-CYC1t, linearized with PstI endonuclease in a molar excess of 10:1, and transformed into trp1-Δ yeast strain by the standard method with lithium acetate [33]. The yeast transformants were grown on a selective medium containing no uracil. Resulting yeast colonies were grown in liquid selective medium without uracil. Plasmid DNA was isolated from yeast cultures by vortexing with glass beads (SigmaAldrich, St. Louis, MO, USA, #G8772) and a phenol/chloroform mixture using a Precellys 24 device (Bertin Technologies, Montigny-le-Bretonneux, France), with subsequent purification performed using the GeneJet Plasmid Purification kit (Thermo Fisher Scientific #K0503, Waltham, MA, USA) and precipitation performed using NaOAc/EtOH. Purified plasmids were used to transform chemocompetent *E. coli* cells. Then, the assembled plasmids were purified from *E. coli* colonies and verified by restriction analysis and Sanger sequencing.

### 4.4. sgRNA Expressing Plasmids

We used RNA spacers against *ADE2* gene described in [19]. Briefly, long oligonucleotides encoding spacers and flanking regions (ADE2-lit-RPR-F, ADE2-2-RPR, ADE2-3-RPR, ADE2-lit-NGA, ADE2-lit-NAG, ADE2-3-NGA, and ADE2-3-NAG) were cloned into the pRPR1_gRNA_handle_RPR1t vector [34] and cut at the HindIII site using the Gibson Assembly Master Mix (NEB #E2611, Ipswich, MA, USA). The reaction mixture was transformed into *E. coli* chemocompetent cells. The assembled plasmids were purified from *E. coli* colonies and verified by Sanger sequencing. Sequences of oligonucleotides are given in Appendix A.

### 4.5. Measurement of SpyCas9 Activity

The yeast strain trp1-Δ was cotransformed with mix of 1 µg Cas9-expressing plasmid encoding Trp1p as a selection marker, 1 µg sgRNA-expressing plasmid encoding Leu2p as a selection marker, a spacer against *ADE2*, and 0.5 µg of double-stranded oligonucleotide as a donor for recombination carrying a TAG stop codon in the *ADE2* frame and mismatches in the PAM in the target region of the *ADE2*. Integration of the donor oligonucleotide into the target locus should prevent the expression of the enzyme and further target cleavage by the CRISPR/Cas9 system. The transformed yeast strains were grown on a selective medium lacking tryptophan and leucine with a low concentration of adenine (1 µg/mL) for 7 days at 30 °C for development of red pigmentation due to *ADE2* gene dysfunction [35]. Then, the numbers of red and white colonies were counted manually or using the Clono-Counter program [36].

### 4.6. Molecular Dynamics (MD) Simulations

Molecular dynamics (MD) simulations were carried out using the GROMACS 2023.2 software [37]. The initial model, 6o0y [31], was employed as a starting point. As certain segments of the peptide chain and DNA were missing in the enzyme’s structure, they were reconstructed using the AlphaFold algorithm [38] in the ChimeraX v 1.6 program [39]. Additionally, Mg^2+^ ions were manually added in positions corresponding to those outlined in reference [40]. This process led to the geometry of the wild-type SpyCas9, which served as the reference structure for subsequent analysis. Before generating models of SpyCas9 Sniper variants, a preparatory stage was performed, involving meticulous optimization of the wild-type SpyCas9 variant’s geometry through multiple steps. Energy minimization using the steep and L-BFGS algorithms was interleaved with multi-stage equilibrations over a 5 ns duration at 300 K. This preparation continued until all artifacts arising from prohibitively close atomic nuclei positions were completely eliminated in the resulting structure. The geometry obtained during the preparatory phase was used as the starting point for MD simulations of the wild-type SpyCas9 and served as the basis for generating all SpyCas9 variants investigated in this study. MD simulations for each model were conducted following a unified protocol (https://github.com/intbio/gmx_protocols, accessed 9 October 2023) typical for protein-nucleic acid complex computations. The simulations employed the amberff19SB (apoenzyme), amberOL21 (DNA), and amberOL3 (RNA) force fields [41], with the system enclosed in a cubic box sized to ensure that the distance from the box edge to the nearest molecule atom was not less than 1 nm. The TIP3P water model was used, and the system was neutralized by Na^+^ ions. The initial stage involved energy minimization of the geometry using steep minimization up to a maximum force of 500 kJ/mol·nm with a 0.01 nm minimization step. Further optimization was conducted using the L-BFGS method. Subsequently, the system was subjected to equilibration across several stages with varying parameters. All optimized and equilibrated geometries were assessed for structural integrity and stability before initiating MD simulations. MD simulations were carried out for 30 ns for each of the seven geometries at a temperature of 300 K. To evaluate the results of the MD calculations, root-mean-square deviation (RMSD) analysis was performed in three variations:RMSD analysis of protein geometries concerning their initial structures (at the beginning of MD simulations).RMSD analysis of the positions of amino acids and Mg^2+^ in the HNH active center.RMSD analysis of the positions of amino acids and Mg^2+^ in the RuvC active center.

Amino acid residues were included in the respective centers if they were in proximity to the coordinated Mg^2+^ at a distance of no more than 5 Å.

## Figures and Tables

**Figure 1 ijms-25-00444-f001:**
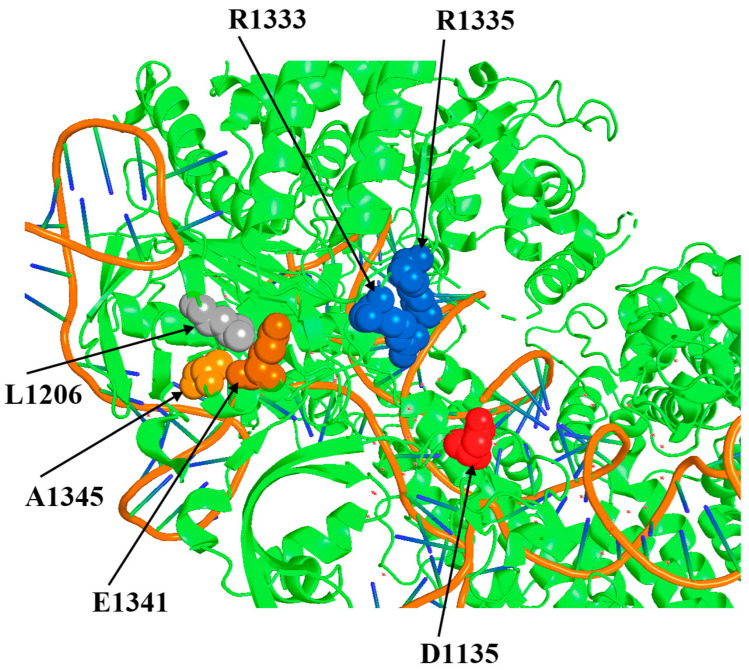
Positions around the PAM-recognizing residues R1333 and 1335 that can influence the PID tertiary structure and SpyCas9 activity.

**Figure 2 ijms-25-00444-f002:**
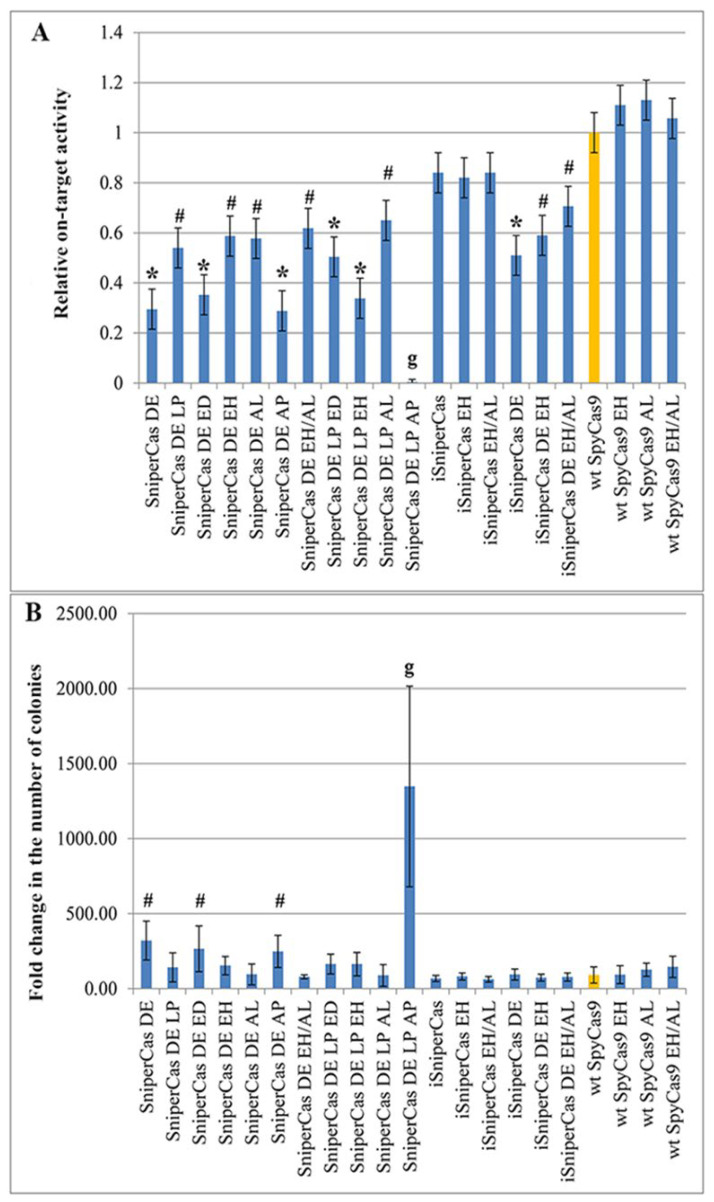
(**A**) Activity of different SpyCas9 variants carrying mutations E1341H and A1345L in comparison with parent forms (SniperCas DE, iSniperCas, iSniperCas DE, and wt SpyCas9). Relative on-target activity of SpyCas9 variants was calculated as the ratio of red colonies to white colonies and then normalized to the activity of wt SpyCas9 (indicated by orange), which was set as 1. (**B**) The number of yeast colonies after transformation by SpyCas9 variants anticorrelates with the activity of SpyCas9 variants. Results were obtained with *ADE2*-lit spacer. Statistical significance: # denotes *p* < 0.05 according to the Student’s *t*-test; * denotes *p* < 0.01 according to the Student’s *t*-test; **g** denotes *p* < 0.001 according to Welch’s *t*-test.

**Figure 3 ijms-25-00444-f003:**
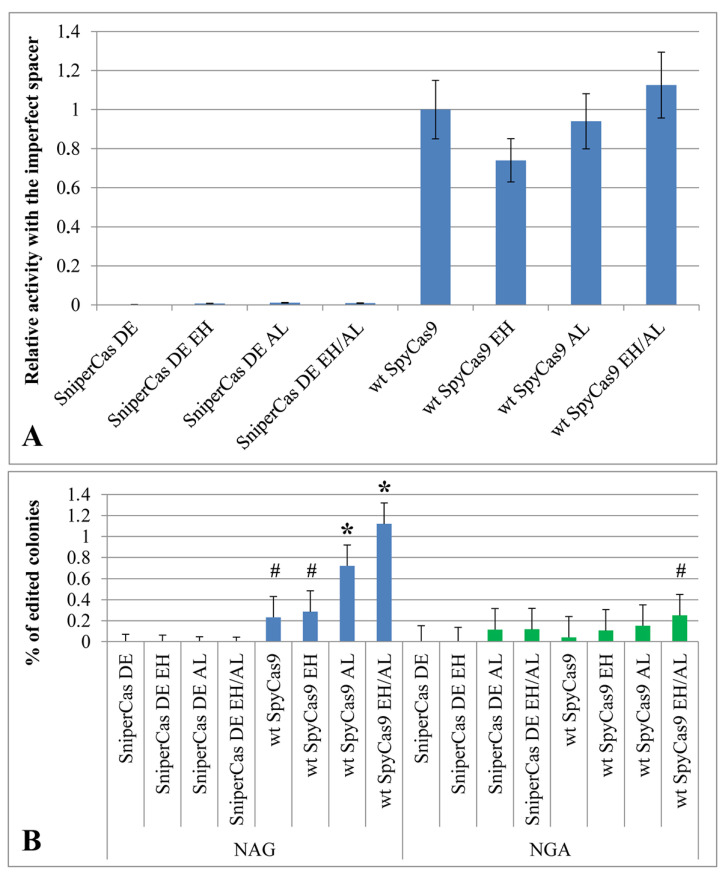
(**A**) The specificity of SpyCas9-carrying mutations E1341H and A1345L evaluated in the *ADE2* test system using the AL11 imperfect spacer. The effectiveness of editing relative to wt SpyCas9 is shown. The low number of edited colonies indicates the high specificity of all Sniper series modifications. (**B**) The specificity of SpyCas9 variants with mutations E1341H and A1345L evaluated using spacers with alternative NGA and NAG PAMs. Statistical significance: # *p* < 0.05 according to the Student’s *t*-test; * *p* < 0.01 according to the Student’s *t*-test.

**Figure 4 ijms-25-00444-f004:**
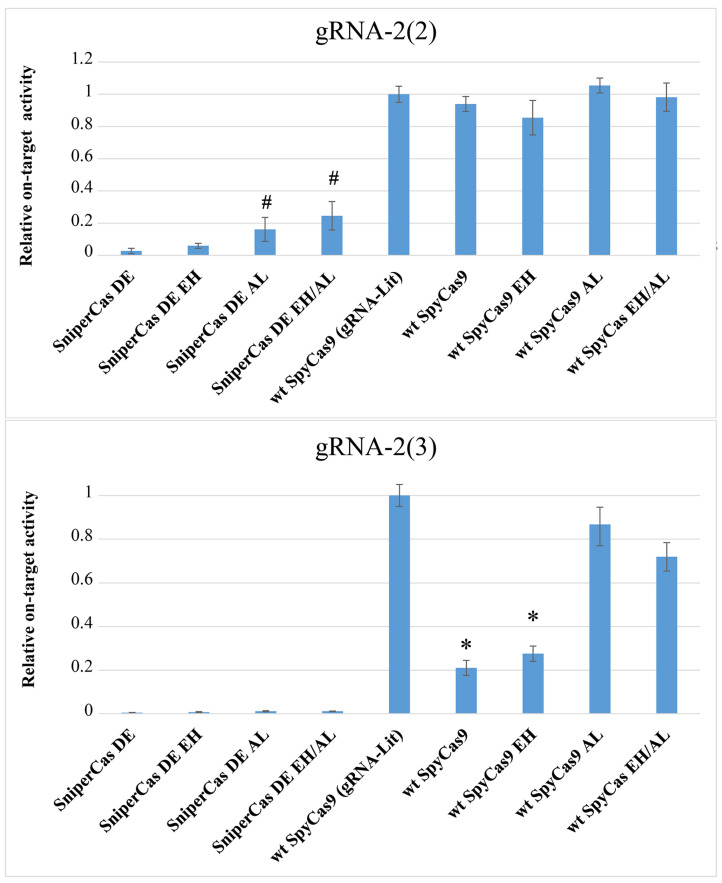
Nucleosome effect on SpyCas9 variants’ relative on-target activity evaluated using alternative spacers 2(2) and 2(3). The activity of wild-type SpyCas9 using gRNA-lit was taken as a unit. The activity of wild-type SpyCas9 is set to 100%. Statistical significance: # *p* < 0.05 when compared to wt SpyCas9 (gRNA-Lit) according to the Student’s *t*-test; * *p* < 0.01 when compared to wt SpyCas9 (gRNA-Lit) according to the Student’s *t*-test.

**Figure 5 ijms-25-00444-f005:**
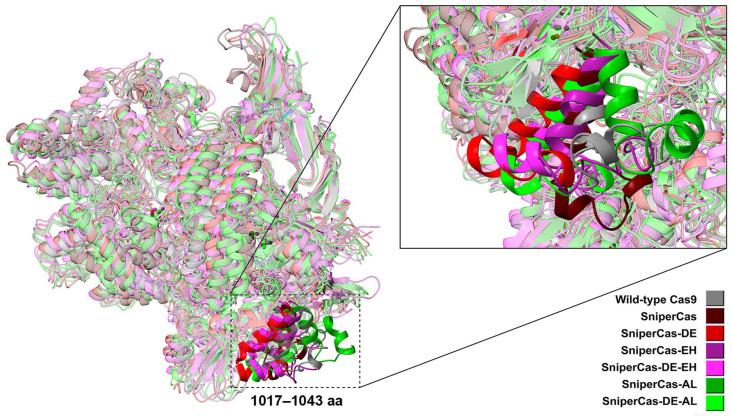
Geometries of SniperCas variants relative to wild-type SpyCas9 at the final point of MD simulations.

**Figure 6 ijms-25-00444-f006:**
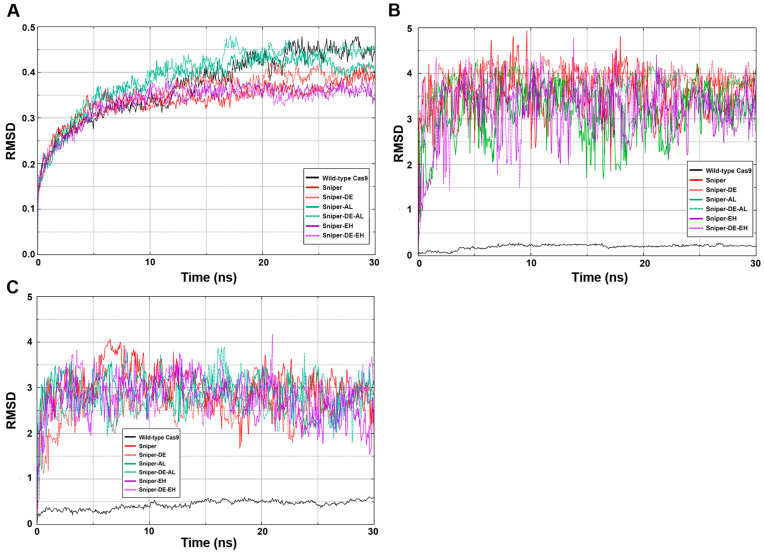
RMSD analysis results. (**A**) SpyCas9 full-length proteins. (**B**) Residues of HNH catalytic domain. (**C**) Residues of RuvC catalytic domain.

**Figure 7 ijms-25-00444-f007:**
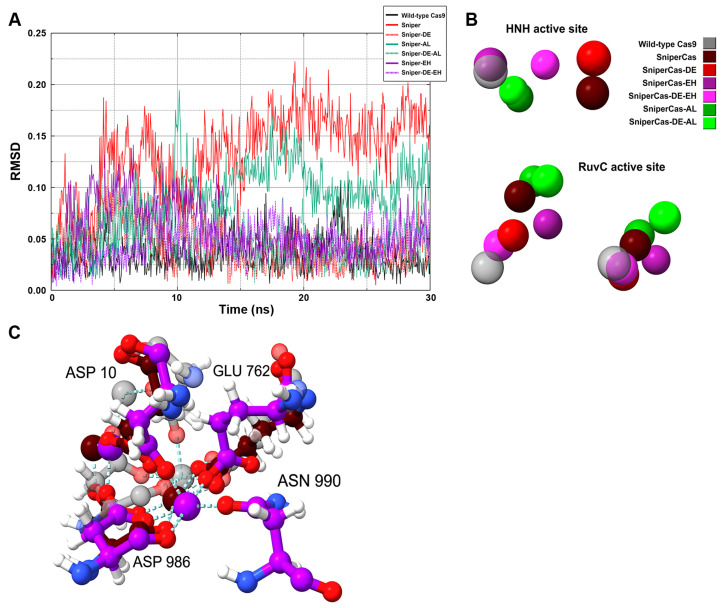
Analysis of the active sites of SpyCas9 variants. (**A**) RMSD analysis of the position of Mg^2+^ ions in the HNH and RuvC active sites of SpyCas9. (**B**) Relative positions of Mg^2+^ ions in SpyCas9 active sites. All geometries are pre-aligned relative to the wild-type SpyCas9. (**C**) Coordination of Mg^2+^ ions in the RuvC active site. The wild-type SpyCas9 is depicted in gray, the SniperCas variant in maroon, and SniperCas-EH in violet.

**Table 1 ijms-25-00444-t001:** SpyCas9 variants used in this work.

	Cas9 Variant	Mutations
1	wt SpyCas9	-
2	wt SpyCas9 EH	E1341H
3	wt SpyCas9 AL	A1345L
4	wt SpyCas9 EH/AL	E1341H, A1345L
5	SniperCas DE	F539S, M763I, K890N, D1135E
6	SniperCas DE ED	F539S, M763I, K890N, D1135E, E1341D
7	SniperCas DE EH	F539S, M763I, K890N, D1135E, E1341H
8	SniperCas DE AL	F539S, M763I, K890N, D1135E, A1345L
9	SniperCas DE AP	F539S, M763I, K890N, D1135E, A1345P
10	SniperCas DE EH/AL	F539S, M763I, K890N, D1135E, E1341H, A1345L
11	SniperCas DE LP	F539S, M763I, K890N, D1135E, L1206P
12	SniperCas DE LP ED	F539S, M763I, K890N, D1135E, L1206P, E1341D
13	SniperCas DE LP EH	F539S, M763I, K890N, D1135E, L1206P, E1341H
14	SniperCas DE LP AL	F539S, M763I, K890N, D1135E, L1206P, A1345L
15	SniperCas DE LP AP	F539S, M763I, K890N, D1135E, L1206P, A1345P
16	iSniperCas	D147Y, P411T, F539S, M763I, K890N
17	iSniperCas EH	D147Y, P411T, F539S, M763I, K890N, E1341H
18	iSniperCas EH/AL	D147Y, P411T, F539S, M763I, K890N, E1341H, A1345L
19	iSniperCas DE	D147Y, P411T, F539S, M763I, K890N, D1135E
20	iSniperCas DE EH	D147Y, P411T, F539S, M763I, K890N, D1135E, E1341H
21	iSniperCas DE EH/AL	D147Y, P411T, F539S, M763I, K890N, D1135E, E1341H, A1345L

**Table 2 ijms-25-00444-t002:** SpyCas9 variants used for MD simulations.

	Name	Amino Acid Substitution
1	WT SpyCas9	-
2	SniperCas	F539S, M763I, K890N
3	SniperCas DE	F539S, M763I, K890N, D1135E
4	SniperCas AL	F539S, M763I, K890N, A1345L
5	SniperCas EH	F539S, M763I, K890N, E1341H
6	SniperCas DE AL	F539S, M763I, K890N, D1135E, A1345L
7	SniperCas DE EH	F539S, M763I, K890N, D1135E, E1341H

## Data Availability

The datasets generated during and/or analyzed during the current study are available from the corresponding author upon reasonable request.

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
