# Peer review of "Increasing the Activity of the High-Fidelity SpyCas9 Form in Yeast by Directed Mutagenesis of the PAM-Interacting Domain"

_ijms, 2023, doi:10.3390/ijms25010444_

Round 1
Reviewer 1 Report
Comments and Suggestions for Authors
The manuscript “Increasing the Activity of the High-Fidelity SpyCas9 Form in Yeast by Directed Mutagenesis of the PAM Interacting Domain” by Artem I. Davletshin aims in an optimization of SpyCas9 nuclease for genome editing. The idea of Cas9 optimization in well-known and the main goal in this area is to increase a specificity of cleavage, at the same time not compromising the activity. The latter is especially relevant for condensed chromatin regions since tight interaction with nucleosome impeded DNA cleavage.
In this work authors used hyperprecise SniperCas9 as a starting point to introduce mutations that would increase the activity without compromising selectivity. A number of mutations have been introduced to SniperCas9 and SpyCas9 apart and in combinations aiming at an increase in the activity while retaining a selectivity. The mutant forms were tested of activity and selectivity towards protospacers carrying a substitution in the region complementary to sgRNA as well as in the PAM. In addition, the ability to cleave nucleosome-packed yeast DNA was likewise tested.
The results are of value to the community since the goals have been achieved, albeit not ideally. Moderate editing of the manuscript text would benefit the clarity of the presentation as detailed below.
1. l.36 …off-target activity (the ability to cause mutations in random genome regions)
I would suggest changing to “off-target activity (the ability to cause mutations in genome regions partially matching the sgRNA)
2. l. 51 … all CRISPR/Cas9 systems that hydrolyze DNA require the presence of a PAM (Protospacer Adjacent Motif) with NGG sequence…
This is not correct, there are Cas nucleases with more diverse PAM sites.
3. l.84 The rationale for obtaining either mutation is completely unclear. Was it rational design of screening of any kind? Was D1135E mutation present in the gene before L1206P being introduced? Presumably yes, since it "compensate". Then, how was it introduced and for what reason? Authors should not assume a reader to know all pre-history of the particular Cas protein variant among hundreds of variants published in recent years.
4. l.90 …E1341D as a replacement by a neutral residue…
probably “…replacement by a similarly charged amino acid”?
5. l.94 …A1345P similar to L1206P… Are those substitutions similar since both of them substitute an amino acid to proline? Similarity in mutations usually have some meaning beyond amino acid identity, e.g. equivalence of structural context or function.
Author Response
We are very grateful to the reviewer for his careful reading of the manuscript and valuable comments that helped us to improve it.
- l.36 …off-target activity (the ability to cause mutations in random genome regions)
I would suggest changing to “off-target activity (the ability to cause mutations in genome regions partially matching the sgRNA)
Response. We are agree and changed this sentence.
- l. 51 … all CRISPR/Cas9 systems that hydrolyze DNA require the presence of a PAM (Protospacer Adjacent Motif) with NGG sequence…
This is not correct, there are Cas nucleases with more diverse PAM sites.
Response. – Thank you very much for the comment. We completely agree that there are Cas proteins that recognize different PAMs or do not require a PAM sequence at all. In this sentence we were referring to the CRISPR/Cas9 system from Streptococcus pyogenes, so we corrected the sentence.
- l.84 The rationale for obtaining either mutation is completely unclear. Was it rational design of screening of any kind? Was D1135E mutation present in the gene before L1206P being introduced? Presumably yes, since it "compensate". Then, how was it introduced and for what reason? Authors should not assume a reader to know all pre-history of the particular Cas protein variant among hundreds of variants published in recent years.
Response. – We have added a more detailed description of work on modifications to the PAM of the SpyCas9 recognition domain to optimize its properties.
- l.90 …E1341D as a replacement by a neutral residue…probably “…replacement by a similarly charged amino acid”?
Response. – Thank You, we've corrected that unfortunate mistake.
- l.94 …A1345P similar to L1206P… Are those substitutions similar since both of them substitute an amino acid to proline? Similarity in mutations usually have some meaning beyond amino acid identity, e.g. equivalence of structural context or function.
Response. – The analogy is the replacement of a relatively free in mobility alanine with a proline residue, increasing the rigidity of the polypeptide chain site and consequently changing the structure of the PAM-binding domain, similar to the L1206P substitution. A corresponding explanation has been added to the text of the revised version of the manuscript.
Reviewer 2 Report
Comments and Suggestions for Authors
This well written manuscript describes advances in the development of SpyCas9 with improved activity and specificity for editing the yeast genome. The authors use molecular modeling to identify candidate regions within the protospacer adjacent motif to target and create amino acid substitutions using directed mutagenesis. They test the on-target specificity and the activity of their constructs to determine their effectiveness. The context of their work is well described as are their experimental procedures and results. Although the study is restricted to S. cerevisiae it should facilitate work on this important organism and likely has broader biotechnological applications.
Additional comments:
This manuscript follows a similar study by essentially the same group in which they used a combination of rational design and random mutagenesis to improve on-target activity of high-fidelity Cas9 editors (ref. 19). As in the current study, the previous one noted that D147Y and P411T mutations increase the activity of Cas9 and that L1206P mutations weaken the chromatin barrier for Cas9. The novelty, and thus the strength, of the current study stems from the use of a rational approach, using molecular modeling based on a crystalline structure to identify amino acid residues around the PAM recognition site as potential targets for mutation with the goal of increasing activity and specificity. They created a series of mutations and tested their specificity and activity using a phenotypic technique and characterized their active sites using RMSD analysis. Overall, the study represents a significant, although incremental advance in the area. However, there are several points that should be addressed:
Figure 2. It is not clear what “g” signifies in this figure.
Figure 3. Yeast colonies edited for ADE2 are reported only as the percentage of (edited) red colonies. The percentage of colonies with sectoral growth (if any) should be noted.
It is unclear what the gold standard for an improved Cas9 editor is in this manuscript. The authors normalize to SpyCas9 and make their statistical comparisons to the activity of iSniperCas DE but fail to provide a rationale for the use of this standard.
Section 4.2. More details should be provided regarding the statistical analysis and how it was used. For example, were the samples normally distributed? Did they have the same variance?
The authors raise the issue of off-target CRISPER/Cas editing and apparently measure specificity of their mutants based on “on-target” activity as in Figure 4. However, they fail to enumerate off-target events.
The English grammar in this manuscript needs moderate editing for clarity.
Author Response
We are very grateful to the reviewer for his careful reading of the manuscript and valuable comments that helped us to improve it.
Figure 2. It is not clear what “g” signifies in this figure.
Response. g denotes statistical significance at the level of less than 0.001 (as well as # is p < 0.05 and * is p < 0.01). In the figure caption, the statistical significance of differences is indicated more clearly.
Figure 3. Yeast colonies edited for ADE2 are reported only as the percentage of (edited) red colonies. The percentage of colonies with sectoral growth (if any) should be noted.
Response. Chimeric (sectoral) colonies are sometimes present in all experiments. Roughly, less active forms give a slightly higher % of chimeric colonies. Accordingly, the lowest number of chimeras (~ 1%) in our collection of mutant forms of Cas9 was produced by wt SpyCas9 and iSniperCas, the other mutants produced between 2 and 5% chimeras in different experiments, with no significant difference between the original SniperDE modification and its improved variants. Therefore, we do not present data on chimerism, preferring the counting of fully edited colonies, the percentage of which for the different Cas9 forms studied ranges from 20 to 80% with a reliable difference.
It is unclear what the gold standard for an improved Cas9 editor is in this manuscript. The authors normalize to SpyCas9 and make their statistical comparisons to the activity of iSniperCas DE but fail to provide a rationale for the use of this standard.
Response. We have revised Figures 2 and 3 so that the activity of all modified forms of SpyCas9 is given relative to wt SpyCas9, which was set as 1.
Section 4.2. More details should be provided regarding the statistical analysis and how it was used. For example, were the samples normally distributed? Did they have the same variance?
Response. We used additional tests to assess the normality of the sampling distribution and to assess differences in variance. The statistical significance of differences between the mean of two compared independent samples was recalculated using appropriate t-tests. Section 4.2. describes the statistical analysis performed in more detail.
The authors raise the issue of off-target CRISPER/Cas editing and apparently measure specificity of their mutants based on “on-target” activity as in Figure 4. However, they fail to enumerate off-target events.
Response. Because we used a haploid strain of S. cerevisiae, and S. cerevisiae has a very weak NHEJ repair system, off-target editing results in chromosomal breaks that cannot be efficiently repaired by highly active homologous recombination in the absence of an appropriate template. Therefore, we developed a model to detect off-target editing of the CRISPR/Cas9 system using the ADE2 gene. Such a model is based on a guide RNA with an imperfect spacer against ADE2, which carries a nucleotide substitution at the fifth position from the 5'-end of the spacer, and the corresponding DNA repair template. Thus, in the case of off-target activity, the Cas9 variant will utilize this imperfect sgRNA and edit ADE2, whereas the high-fidelity Cas9 variant will not be able to use this imperfect sgRNA and therefore will not edit ADE2. Thus, we know exactly the off-target site.